# Curcumin for Inflammation Control in Individuals with Type 2 Diabetes Mellitus and Metabolic Dysfunction-Associated Steatotic Liver Disease: A Randomized Controlled Trial

**DOI:** 10.3390/nu17121972

**Published:** 2025-06-10

**Authors:** Metha Yaikwawong, Laddawan Jansarikit, Siwanon Jirawatnotai, Somlak Chuengsamarn

**Affiliations:** 1Department of Pharmacology, Faculty of Medicine Siriraj Hospital, Mahidol University, Bangkok 10700, Thailand; metha.yai@mahidol.ac.th (M.Y.); laddawan.jas@mahidol.ac.th (L.J.); siwanon.jir@mahidol.ac.th (S.J.); 2Siriraj Center of Research Excellence for Precision Medicine and Systems Pharmacology, Faculty of Medicine Siriraj Hospital, Mahidol University, Bangkok 10700, Thailand; 3Faculty of Pharmacy, Silpakorn University, Nakhon Prathom 73000, Thailand; 4Division of Endocrinology and Metabolism, Faculty of Medicine, HRH Princess Maha Chakri Sirindhorn Medical Center, Srinakharinwirot University, Nakhon Nayok 26120, Thailand

**Keywords:** curcumin, metabolic dysfunction-associated steatotic liver disease (MASLD), nutritional supplements, type 2 diabetes

## Abstract

**Background**: Curcumin, a bioactive polyphenol derived from turmeric, has demonstrated potential therapeutic effects in metabolic dysfunction-associated steatotic liver disease (MASLD) by modulating inflammation, oxidative stress, hepatic fat accumulation, and fibrosis. **Objective**: To evaluate the efficacy of curcumin in reducing hepatic steatosis and liver stiffness in patients with MASLD. **Methods**: In this randomized, double-blind, placebo-controlled trial, 78 patients with type 2 diabetes mellitus (T2DM) and MASLD were randomly assigned to receive either curcumin (1500 mg/day) or placebo for 12 months. The primary outcome was the change in tumor necrosis factor (TNF) levels. Secondary outcomes included changes in interleukin-1 beta (IL-1β), interleukin-6 (IL-6), antioxidant enzyme activities (glutathione peroxidase, superoxide dismutase), the oxidative stress marker malondialdehyde, non-esterified fatty acids, and hepatic parameters (hepatic steatosis and liver stiffness). Assessments were conducted at baseline and at 3, 6, 9, and 12 months. **Results**: All participants completed the study (curcumin group: *n* = 39; placebo group: *n* = 39). Curcumin significantly reduced TNF levels at all follow-up points compared to placebo (*p* < 0.001). IL-1β, IL-6, and malondialdehyde levels also declined significantly (*p* < 0.001), while antioxidant enzyme activities, including glutathione peroxidase and superoxide dismutase, increased significantly (*p* < 0.001), indicating improved oxidative balance. Furthermore, curcumin led to significant reductions in non-esterified fatty acids, total body fat, BMI, hepatic steatosis, and liver stiffness compared to placebo. **Conclusions**: Twelve months of curcumin supplementation improved glycemic control, reduced systemic inflammation and oxidative stress, and significantly improved hepatic steatosis and liver stiffness in patients with MASLD. These findings support curcumin as a promising adjunctive therapy for MASLD management.

## 1. Introduction

Metabolic dysfunction-associated steatotic liver disease (MASLD) is defined by hepatic steatosis in conjunction with at least one condition, such as obesity, overweight status, type 2 diabetes mellitus (T2DM), or laboratory markers of metabolic dysregulation [1]. The term MASLD was introduced in 2023 by an international expert panel as a replacement for nonalcoholic fatty liver disease. Its incidence has surged alongside the global increase in obesity, T2DM, and metabolic syndrome, making MASLD the most prevalent chronic liver disease worldwide [2,3]. It is also a growing contributor to severe liver conditions, including cirrhosis, liver failure, and hepatocellular carcinoma [4].

The pathogenesis of MASLD is multifactorial, involving both environmental influences and genetic predispositions [5]. Hepatic steatosis develops primarily from the accumulation of free fatty acids, which originate from systemic circulation and dietary sources. This process is further exacerbated by de novo hepatic lipogenesis. Disease progression features hepatic stress responses, oxidative stress, mitochondrial dysfunction, stellate cell activation, and alterations in the intestinal microbiota, all of which perpetuate liver damage [5]. Historically, liver fibrosis severity has been gauged through liver biopsy, an invasive method prone to sampling errors. Emerging noninvasive imaging techniques, such as magnetic resonance elastography and transient elastography (FibroScan), are increasingly used to measure liver stiffness, a key indicator of fibrosis severity.

Curcumin, the main curcuminoid in turmeric (*Curcuma longa*), has well-documented antioxidant and anti-inflammatory properties [6,7,8]. In vitro and in vivo studies show that curcumin can inhibit hepatic stellate cell activation by blocking leptin signaling, regulating glucose and lipid metabolism, and maintaining the balance of extracellular matrix synthesis and degradation [9]. Various animal models have demonstrated their anti-inflammatory, antifibrotic, and antihyperlipidemic effects [10,11].

Several randomized clinical trials have demonstrated that curcumin possesses hepatoprotective, antioxidant, anti-inflammatory, antidiabetic, and lipid-lowering properties in humans. Notably, most of these studies were of relatively short duration, typically around 8 weeks. For instance, Rahmani et al. (2016) conducted an 8-week randomized, double-blind, placebo-controlled trial in patients with non-alcoholic fatty liver disease (NAFLD), finding that curcumin supplementation significantly reduced liver fat content and improved metabolic parameters compared to placebo [12]. Similarly, Panahi et al. (2017) reported that 8 weeks of phytosomal curcumin supplementation led to significant improvements in liver enzymes and ultrasonographic findings in NAFLD patients [13]. These findings suggest that curcumin may be beneficial in managing NAFLD.

Despite these promising findings, the number of randomized clinical trials on curcumin supplementation remains limited. This study aimed to evaluate the efficacy of curcumin on inflammatory and hepatic parameters in patients with MASLD. It also assessed curcumin’s antioxidant and weight management effects through a double-blind, placebo-controlled design to determine its potential as a therapeutic intervention.

## 2. Methods

### 2.1. Study Design and Participants

This randomized, double-blind, placebo-controlled trial was conducted at the HRH Princess Maha Chakri Sirindhorn Medical Center, Srinakharinwirot University, Nakhon Nayok, Thailand. Data analysis and reporting followed the Consolidated Standards of Reporting Trials guidelines [14] (see Appendix A). Seventy-eight patients with MASLD were recruited from a cohort of individuals with T2DM based on predefined inclusion and exclusion criteria. A detailed enrollment flow chart is provided in Appendix A.

### 2.2. Inclusion Criteria

Participants were eligible if they were at least 35 years old and had hepatic steatosis confirmed by FibroScan (Echosens, Paris, France) with a controlled attenuation parameter (CAP) greater than 248 dB/m [15]. Additional criteria included no history of significant alcohol consumption (fewer than 20 g/day for women and fewer than 30 g/day for men) and no other liver disorders, malignancies, or kidney disorders. Individuals with a recent history of substantial weight loss or bariatric surgery were also excluded. All participants received metformin for hyperglycemia management, and other antidiabetic medications were excluded at recruitment to reduce potential confounding effects.

### 2.3. Exclusion Criteria

Exclusion criteria consisted of pregnancy or breastfeeding, participation in another clinical trial, hypersensitivity to the study supplement, uncontrolled hypertension, uncontrolled dyslipidemia, or unwillingness to continue the trial. Individuals with T2DM who required antidiabetic medications other than metformin were excluded, and no participants received insulin injections. Antidiabetic treatment regimens were not adjusted during the study. Hypertension and dyslipidemia were managed with stable antihypertensive and antidyslipidemic medications, respectively, with no changes permitted. Details on specific medications are provided in Table 1.

### 2.4. Study Protocol

The study design featured a 12-month intervention preceded by a 3-month run-in period (month −3 to month 0). During the run-in, all participants received standardized education on diet and exercise. They were also given written lifestyle recommendations and attended a 20–30 min one-on-one workshop emphasizing healthy lifestyle habits. After this baseline phase, participants underwent the 12-month intervention. They were advised to follow medical nutrition therapy guidelines and engage in regular physical activity. Metformin was administered throughout the study to manage hyperglycemia.

### 2.5. Ethics

Ethical approval was obtained from the Faculty of Medicine, Srinakharinwirot University, Bangkok, Thailand (SWUEC/FB 4/2556). The trial was registered with the Thai Clinical Trials Registry (TCTR20140303003) and conducted in accordance with the Declaration of Helsinki. All participants provided written informed consent prior to enrollment

### 2.6. Dietary and Exercise Recommendations

Before the study, participants were counseled on dietary and exercise guidelines that emphasized consuming low-glycemic index foods, such as legumes and whole grains, and increasing dietary fiber. They were also instructed to engage in at least 150 min of moderate-intensity aerobic exercise per week [16,17]. Participants were encouraged to maintain consistent physical activity throughout the trial. To assess dietary intake, they completed a questionnaire at baseline and again at 12 weeks. The questionnaires were analyzed using the Computer Dietary Guidance System Software (version 3.0; Peking Union Medical College & West China Center of Medical Sciences, Beijing, China). A baseline questionnaire further evaluated dietary habits, including the frequency and quantity of meat, milk, egg, and vegetable consumption (Appendix A).

### 2.7. Randomization Procedures

Following participant screening, informed consent, and completion of diet and lifestyle training, a random allocation sequence was generated using computer-based methods or random number tables. The sequence was concealed from individuals responsible for enrolling participants to prevent selection bias. Participants were then randomly assigned to the intervention or control group according to the concealed allocation sequence. Regular monitoring and verification were conducted to confirm the maintenance of blinding and to ensure consistent CAP and liver stiffness. Comprehensive documentation of the randomization process, allocation sequence, and group assignments was securely maintained for subsequent analysis.

### 2.8. Blinding Procedures

To maintain blinding among participants and healthcare providers, placebo capsules were designed to match curcumin capsules in appearance, taste, and administration method. All participants received their assigned treatments according to a standardized protocol, with blinding preserved throughout the study. Healthcare providers applied uniform procedures for both groups and followed standardized administration protocols, minimizing potential bias. Treatment containers were labeled with codes accessible only to the trial coordinator, and providers were trained to avoid revealing intervention details inadvertently. Regular follow-up assessments monitored outcomes and adverse effects, while compliance checks ensured adherence to trial protocols. All outcome data were systematically collected to minimize loss to follow-up and maintain data completeness. To evaluate the effectiveness of blinding, both participants and investigators were asked to guess the assigned intervention group (curcumin or placebo) at the conclusion of the study. The accuracy of these guesses was then compared to the expected probability of correct guesses under random chance (50%) using a two-sided exact binomial test.

### 2.9. Intervention

Participants consumed three blinded capsules of either curcumin (250 mg of curcuminoids per capsule) or placebo twice daily, for a total of six capsules per day over 12 months. Both curcumin and placebo capsules were manufactured by the Government Pharmaceutical Organization of Thailand. Capsule compliance was evaluated during follow-up visits at 3, 6, 9, and 12 months by recording capsule counts (see Appendix A).

### 2.10. Preparation of Curcuminoid Capsules

Turmeric (*Curcuma longa* Linn.) rhizomes from Kanchanaburi Province, Thailand, were dried, ground, and extracted with ethanol. The resulting semisolid ethanol extract contained oleoresin and curcuminoids. Oleoresin was removed to yield a curcuminoid extract containing 75–85% total curcuminoids. High-performance thin-layer chromatography was employed to determine peak ratios of curcumin, demethoxycurcumin, and bisdemethoxycurcumin. Each capsule contained 250 mg of this curcuminoid extract, which was encapsulated according to Good Manufacturing Practice standards [18]. Extracted chemical fingerprints and detailed compositions are presented in Appendix A.

### 2.11. Study Outcomes

The primary outcome was a significant reduction in TNF levels in the curcumin group compared with the placebo group. The secondary outcomes included changes in CAP and liver stiffness, measured noninvasively by transient elastography (FibroScan 520 Touch; Echosens, Paris, France). To optimize measurement consistency, investigators underwent rigorous training in transient elastography, and standardized protocols were established for patient preparation and positioning. Patients fasted for at least 2 h before the procedure and were positioned supine with the right arm maximally abducted to allow optimal probe placement in the right mid-axillary line. Measurements were taken at the right liver lobe through intercostal spaces, with the probe maintained perpendicular to the liver capsule to ensure reliable readings. These steps enhanced the reproducibility of CAP and liver stiffness assessments.

Additional secondary outcomes included changes in waist circumference, total body fat (TBF), malondialdehyde, superoxide dismutase, glutathione peroxidase, and high-sensitivity C-reactive protein. Adverse effects were defined as elevated creatinine (≥1.2 mg/dL), aspartate transaminase or alanine transaminase levels that were at least three times the upper limit of normal, and any self-reported symptoms [19].

### 2.12. Data Collection and Measurement Methods

Measurements were conducted at baseline and at 3, 6, 9, and 12 months postintervention. Baseline data included demographic information, medical history, medication use, body weight, height, waist circumference, and vital signs. Waist circumference, an indicator of abdominal obesity, was measured horizontally between the iliac crest and the costal margin [20].

Body fat percentage (BFP) was estimated using equations developed by Xin Liu et al. [21], which incorporates body mass index (BMI), age, and waist circumference. The following equations have been validated in independent population-based samples of Chinese men and women, enhancing their applicability to Asian populations. The formulas are as follows:

For men:BFP = 0.567 × BMI + 0.101 × age + 0.23 × waist circumference − 5.4

For women:BFP = 0.439 × BMI + 0.221 × age + 0.18 × waist circumference − 9.4

After calculating BFP, TBF in kilograms was derived using the following equation:TBF = (BFP × total body weight [kg])/100.

Fasting plasma glucose, glycated hemoglobin (HbA1c), aspartate aminotransferase, alanine aminotransferase, and gamma-glutamyl transferase (GGT) were measured using standard procedures. Plasma triglyceride (TG) levels were analyzed with diagnostic kits (Randox Laboratories Ltd., Antrim, UK) on an automated analyzer (spACE; Schiapparelli Biosystems Inc., Columbia, MD, USA).

Serum levels of pro-inflammatory cytokines—interleukin-1 beta (IL-1β), interleukin-6 (IL-6), and tumor necrosis factor (TNF)—were successfully measured using validated enzyme-linked immunosorbent assays (ELISAs). Blood samples were collected, processed, and stored according to standard protocols to ensure sample integrity. IL-1β was measured using the Human IL-1β ELISA Kit (Abcam, Cambridge, UK, catalog No. ab214025), which has a sensitivity of 14.06 pg/mL and intra- and inter-assay coefficients of variation (CVs) of 4.8% and 5.6%, respectively. IL-6 was assessed using the Human IL-6 ELISA Kit (Abcam, Cambridge, UK, catalog No. ab178013), with a sensitivity of 2 pg/mL and CVs of 4.2% (intra-assay) and 6.4% (inter-assay). TNF levels were determined using the Human TNF ELISA Kit (Abcam, Cambridge, UK, catalog No. ab181421), with a sensitivity of 4.32 pg/mL and intra- and inter-assay CVs of 2.5% and 3.1%, respectively.

Total serum antioxidant status was determined by an automated method developed by Erel, which evaluates antioxidative capacity against hydroxyl radical-initiated reactions [22]. Superoxide dismutase and glutathione peroxidase activities were measured calorimetrically with RANSOD and RANSEL kits (Randox Laboratories Ltd., Crumlin, UK) on an Abbott Alcyon 300 automated analyzer (Abbott Laboratories, Abbott Park, IL, USA). Malondialdehyde, a marker of lipid peroxidation and oxidative stress, was quantified by reacting with thiobarbituric acid to form a fluorescent complex; fluorescence intensity was measured at 547 nm (excitation 525 nm) using a Kontron SFM 25A spectrofluorometer (Kontron, Milan, Italy) [23]. Non-esterified fatty acids (NEFA) were measured enzymatically using a commercial kit (Wako Chemicals GmbH, Neuss, Germany).

Lipid accumulation product (LAP) was calculated by the following formulas [24]:

For men:LAP = (waist circumference − 65) × triglycerides (TG) [mmol/L]

For women:LAP = (waist circumference − 58) × TG [mmol/L]

Hepatic steatosis index (HSI) was computed using this formula [25]:HSI = 8 × (ALT/AST ratio) + BMI (+ 2 for women).

Finally, the fatty liver index (FLI) was determined using the following equation [26]:e^0.953×ln(TG)+0.139×BMI+0.718×ln(GGT)+0.053×WC−15.745^/1 + e^0.953×ln(TG)+0.139×BMI+0.718×ln(GGT)+0.053×WC−15.745^ × 100
where BMI is body mass index, GGT is gamma-glutamyl transferase, TG is triglycerides, and WC is waist circumference.

### 2.13. Sample Size

The initial sample size calculation was based on detecting a 2 pg/mL difference in mean serum TNF levels, with 80% statistical power (β = 0.20), requiring 21 participants per group [27]. While TNF is the primary endpoint, secondary outcomes such as liver stiffness are also critical for evaluating liver fibrosis. Based on prior studies evaluating changes in liver stiffness, a sample size of 34 participants was estimated to provide 90% power to detect a 3.1 kPa reduction in liver stiffness [28]. This underscores the necessity of tailoring sample size calculations to the specific effect sizes and variability expected in the secondary outcome. Sample size calculations were performed using G*Power software (version 3.1) [29].

### 2.14. Statistical Analysis

Baseline demographic data are reported as means (SD) for continuous variables or as numbers (percentages) for categorical variables. Outcome variables are presented as medians (interquartile range) at baseline, 3, 6, and 9 months for both groups. A per-protocol analysis was conducted. Between-group comparisons used the independent samples *t*-test for normally distributed data and the Mann-Whitney *U* test for non-normally distributed data. Categorical variables were analyzed using Fisher’s exact test or chi-square tests. Normality was assessed analytically via the Shapiro-Wilk test.

To assess potential sex-specific responses to curcumin supplementation, we conducted sex-stratified linear regression analyses on the change (Δ) in key outcomes over the 12-month intervention period. Each model was adjusted for baseline values and age to control for potential confounding factors.

The assessment of normality and all other statistical analyses were conducted using R software version 4.3.2 (R Foundation for Statistical Computing, Vienna, Austria) [30].

## 3. Results

### 3.1. Baseline Characteristics

Appendix A illustrates the trial flow chart, and Table 1 presents the baseline characteristics of all 78 randomized participants. There were no significant differences between the two groups at baseline. The intervention outcomes were assessed in several domains, including anti-inflammatory and antioxidant effects, changes in non-esterified fatty acid levels, anthropometric measurements, glycemic control, hepatic steatosis, liver stiffness, and potential adverse effects.

Additionally, Appendix A summarizes the mean daily nutrient intake—covering total energy intake (kcal/day), macronutrient distribution (percentage of energy from carbohydrates, protein, and fat), and fiber intake (g/day)—assessed at baseline and after 12 months using validated dietary questionnaires. No significant between-group differences were observed at either time point, indicating that dietary intake remained stable throughout the study. These findings support the comparability of lifestyle factors between groups and reduce the likelihood that dietary variation confounded the observed effects of the intervention.

### 3.2. Anti-Inflammatory Effects

Compared with placebo, curcumin significantly reduced pro-inflammatory cytokines, including TNF, IL-1β, and IL-6, at 3, 6, 9, and 12 months (Table 2).

### 3.3. Antioxidant Effects

Curcumin significantly increased glutathione peroxidase activity compared with placebo (Table 2). Superoxide dismutase activity was also significantly higher in the curcumin group at 3, 6, 9, and 12 months. In contrast, malondialdehyde, a marker of oxidative stress and a potential contributor to MASLD progression, was significantly lower in the curcumin group than in the placebo group at these same time points (Table 2).

### 3.4. Non-Esterified Fatty Acid Levels

Curcumin significantly reduced NEFA levels—a key driver of hepatic fat accumulation and MASLD pathogenesis—compared to placebo at 6 months (Table 2), suggesting enhanced liver metabolic function and reduced lipotoxicity.

### 3.5. Anthropometric Measurements and Weight Management

Waist circumference and TBF were significantly lower in the curcumin group than in the placebo group at 6, 9, and 12 months. BMI was also significantly reduced in the curcumin group at 12 months (Table 2).

### 3.6. Glycemic Control Effects

The median values of diabetes-related blood chemistries, including HbA1c and fasting plasma glucose, were significantly lower in the curcumin group than in the placebo group at 6, 9, and 12 months (Table 2).

### 3.7. CAP and Liver Stiffness

The CAP was significantly lower in the curcumin group than in the placebo group at 3, 6, 9, and 12 months (Table 2). Liver stiffness was also significantly reduced in the curcumin group compared with the placebo group at all follow-up visits (Table 2).

### 3.8. Fatty Liver-Associated Indicators

The FLI was significantly lower in the curcumin group at 3, 6, 9, and 12 months (Table 2). The LAP was also notably lower in the curcumin group at all time points. In addition, the HSI showed significant reductions in the curcumin group at 3, 6, 9, and 12 months (Table 2).

### 3.9. Sex-Stratified Analyses

Curcumin supplementation significantly improved inflammatory markers, oxidative stress, anthropometric indices, metabolic profiles, and hepatic parameters in both male and female participants. Comprehensive results are presented in the Appendix A.

### 3.10. Blinding Assessment

Both participants and investigators were asked to guess the assigned treatment group at the end of the study. As shown in the Appendix A, the accuracy of these guesses did not significantly differ from random chance. Among participants, 48.7% correctly identified their group allocation (*p* = 0.84), and among investigators, 52.6% guessed correctly (*p* = 0.73). These results suggest that blinding was effectively preserved throughout the trial.

### 3.11. Adverse Effects

To assess the potential adverse effects of curcumin, kidney and liver function tests were conducted (Appendix A). No significant differences were found between the groups in aspartate transaminase, alanine transaminase, or creatinine levels. Additionally, no episodes of hypoglycemia were reported in the curcumin group.

### 3.12. Overall Safety and Compliance

Curcumin extract appeared safe for at least 12 months of use. Capsule consumption rates were similar in both groups, indicating comparable compliance (Appendix A). Consequently, the observed effects do not appear to reflect differential adherence.

## 4. Discussion

MASLD affects approximately 30% of the global population and is a major contributor to cirrhosis and hepatocellular carcinoma [31]. This condition is especially common among overweight or obese individuals, with a global prevalence of nearly 50%. In patients with MASLD and metabolic-associated steatohepatitis, the prevalence of T2DM is about 22.5%. Moreover, up to 80% of individuals with MASLD may develop new-onset T2DM [32,33]. The synergistic relationship between MASLD and T2DM exacerbates adverse outcomes beyond what either condition alone would cause. Insulin resistance, which underlies T2DM, compromises glucose uptake in insulin-sensitive tissues such as the liver and adipose tissue [34]. This disruption in glycolipid homeostasis substantially contributes to MASLD progression [35].

Curcumin, known for its cardiovascular protective effects, is regarded as a promising nutraceutical [36]. This study focused on ethanol-extracted curcumin as an intervention to explore a safe, well-tolerated, and widely accessible option for atherosclerosis prevention in patients with T2DM [37].

This double-blind, placebo-controlled trial used the FibroScan 520 Touch, a noninvasive device that measures hepatic steatosis via the CAP [38]. Our findings revealed that 3 months of curcumin supplementation significantly decreased pro-inflammatory cytokines—TNF, IL-1β, and IL-6—compared with placebo. These results underscore curcumin’s pronounced anti-inflammatory effects. Additionally, curcumin’s antioxidant properties were assessed by measuring glutathione peroxidase and superoxide dismutase activities, as well as malondialdehyde levels, a marker of lipid peroxidation. Markers of abdominal obesity (waist circumference) and overall adiposity (TBF and BMI) were also evaluated.

To ensure accurate interpretation of curcumin’s effects, individuals with uncontrolled T2DM were excluded, as changes in their antihyperglycemic regimens could have confounded the findings. Notably, despite well-controlled T2DM, the placebo group exhibited elevated pro-inflammatory cytokine levels, reflecting the chronic inflammatory milieu characteristic of T2DM. This observation underscores curcumin’s potential for mitigating inflammation and metabolic dysregulation in this patient population.

Curcumin’s anti-inflammatory and antioxidant properties have been widely investigated in both in vitro and in vivo models [39,40,41]. Studies show that curcumin reduces IL-1β [42], IL-6 [43], and TNF [43], key cytokines involved in the pathogenesis of MASLD. Excessive levels of these cytokines exacerbate hepatic inflammation and lipid accumulation, thereby advancing MASLD [44,45]. Furthermore, MASLD is marked by increased oxidative stress, lipid peroxidation, and diminished antioxidant defenses [46]. Lipid peroxidation produces inflammatory mediators, including malondialdehyde, a well-established biomarker of oxidative stress [47].

In our study, a 3-month curcumin intervention significantly reduced TNF, IL-1β, and IL-6 levels in patients with MASLD. These results align with those reported by Saadati et al., who observed lower TNF levels after a 12-week curcumin supplementation in patients with non-alcoholic fatty liver disease [48]. Jazayeri-Tehrani et al. similarly noted decreased TNF and IL-6 levels following 3 months of nanocurcumin administration [49], while Ghaffari et al. documented reduced IL-6 after a 10-week turmeric regimen [50].

Our findings also demonstrated that curcumin supplementation substantially lowered malondialdehyde levels and bolstered antioxidant defenses, as evidenced by increased glutathione peroxidase and superoxide dismutase activities [51]. These observations corroborate the work of Ghaffari et al., who reported elevated total serum antioxidant capacity and diminished malondialdehyde levels after turmeric supplementation in non-alcoholic fatty liver disease patients [50].

Additionally, in vivo studies indicate that curcumin supplementation reduces NEFA levels [52,53]. In our study, curcumin significantly lowered NEFA levels after 6 months of intervention in patients with MASLD. These findings align with those of Li-Xin et al., who reported decreased total serum free fatty acid and malondialdehyde levels in curcumin-treated individuals with T2DM [54]. Elevated NEFAs contribute to hepatic lipid accumulation, promote lipotoxicity, and increase oxidative stress and inflammation in hepatocytes, thereby exacerbating metabolic dysfunction. NEFAs thus play a critical role in the metabolic disturbances that underlie MASLD pathogenesis [55].

Curcumin supplementation also led to substantial reductions in waist circumference, TBF, and BMI. Jazayeri-Tehrani et al. similarly observed reduced waist circumference with nanocurcumin, although they did not find a significant change in BMI [49]. In contrast, our study noted a clear BMI reduction, consistent with a meta-analysis of randomized controlled trials showing significant decreases in body weight and BMI following curcumin supplementation [56]. Although the exact mechanisms remain under investigation, curcumin’s anti-inflammatory properties may contribute to BMI reduction. While acute inflammation can lead to increased energy expenditure and weight loss.

While inflammation is typically associated with increased energy expenditure and weight loss in conditions like cachexia, obesity is characterized by chronic low-grade inflammation that contributes to insulin resistance and metabolic dysregulation, thereby hindering weight loss [57,58]. Curcumin’s anti-inflammatory properties may help restore metabolic balance by improving insulin sensitivity, modulating adipokines, and regulating lipid metabolism [59]. Clinical trials have demonstrated that curcumin supplementation can reduce body weight and BMI, even in settings with controlled lifestyle interventions [59,60].

Furthermore, participants receiving curcumin exhibited significant reductions in HbA1c and fasting plasma glucose at 6, 9, and 12 months compared with placebo. Chronic hyperglycemia activates the NF-κB pathway, leading to the production of pro-inflammatory cytokines such as TNF, IL-1β, and IL-6 [61]. Curcumin effectively suppressed TNF levels and improved glycemic control by lowering fasting glucose and insulin levels, independent of weight loss [62]. These findings suggest that curcumin may serve as a valuable adjunct for weight management and glycemic control in individuals with obesity. However, further studies are needed to clarify its mechanisms and optimize its therapeutic use.

Curcumin’s anti-inflammatory and antioxidant properties may underlie its beneficial metabolic effects, including reductions in waist circumference, TBF, visceral fat, and BMI. These changes could explain the observed decreases in CAP. In our study, 12 months of curcumin supplementation significantly lowered CAP. Similarly, Saadati et al. reported a marked reduction in hepatic steatosis following a 12-week curcumin intervention [48]. Our findings also showed that curcumin improved fatty liver-related indices—HSI, FLI, and LAP—after 3 months of treatment. These results align with those of Cicero et al., who documented improvements in HSI, FLI, and LAP after 56 days of curcumin therapy [63]. Naseri et al. likewise found reductions in these indices following 12 weeks of supplementation [64].

We observed significant reductions in liver stiffness. Although liver stiffness is a widely used non-invasive marker of hepatic fibrosis, it is also sensitive to hepatic inflammation. Therefore, elevated liver stiffness may reflect inflammatory activity rather than true fibrotic changes, and the observed reductions may primarily indicate improvements in inflammation rather than definitive fibrosis regression [65]. Curcumin’s well-established anti-inflammatory properties likely contributed to these changes [66]. However, as liver stiffness is influenced by both fibrosis and inflammation, the precise nature of the improvements remains uncertain without histological confirmation.

Regarding safety, curcumin has been well tolerated in humans at doses up to 8000 mg/day, with no significant toxicity observed [67]. This aligns with the 1500 mg/day dosage in our study, which did not cause severe adverse effects. Although multiple randomized clinical trials have highlighted curcumin’s potential for managing MASLD [48,49,50], most interventions lasted only 12 weeks and lacked rigorous safety evaluations. By extending the follow-up to 12 months and systematically assessing adverse effects, our study addressed these limitations. Our findings suggest that curcumin supplementation effectively attenuates inflammatory and oxidative stress pathways, evidenced by reduced pro-inflammatory cytokines and enhanced antioxidant enzyme activity.

It is noteworthy that our study included a higher proportion of women than is typically reported in MASLD research, a condition generally more prevalent in men [68,69]. This discrepancy may stem from the inclusion of postmenopausal women, who experience hormonal changes which may elevate MASLD prevalence to rates potentially exceeding those in men [70]. Despite the female-dominant cohort, sex-stratified analyses revealed that curcumin supplementation consistently improved inflammatory markers, oxidative stress, anthropometric measures, metabolic parameters, and hepatic steatosis in both sexes. These effects remained significant after adjusting for age, suggesting that curcumin’s therapeutic benefits are not sex-dependent. Nonetheless, future studies with larger, more sex-balanced populations are needed to validate these findings and explore potential sex-specific responses.

This study has several strengths, including an extended follow-up period and comprehensive assessments of inflammatory cytokines and antioxidant markers, both of which are vital for understanding metabolism and MASLD pathophysiology. Nevertheless, several limitations warrant discussion. First, MASLD was not histologically confirmed, as liver biopsies were not performed. While non-invasive methods like FibroScan provide quantitative measures of liver stiffness and steatosis, respectively, they may not fully capture subtle histological changes, potentially leading to misclassification, especially in cases where such changes are minimal yet clinically significant [71]. Second, the sample size was relatively small. Third, the single-dose design precluded analysis of dose–response relationships [72]. Lastly, as a single-center randomized controlled trial, the findings may not be generalizable to other populations and settings.

## 5. Conclusions

Curcumin, a bioactive compound in turmeric, exhibits promising hepatoprotective effects through its anti-inflammatory and antioxidant properties. Our findings support the use of curcumin supplementation as a potential therapeutic strategy for MASLD. Given the lack of FDA-approved treatments specifically for MASLD, incorporating curcumin into clinical management may offer a safe and effective alternative for disease prevention and treatment.

## Figures and Tables

**Table 1 nutrients-17-01972-t001:** Baseline characteristics of study participants.

Variable	Placebo	Curcumin	*p* Value *
	Mean ± SD (*n* = 39)	Mean ± SD (*n* = 39)	
Sex (M:F ratio)	14/25 (0.56)	11/28 (0.39)	0.62 ^†^
Age	60.28 ± 9.49	57.33 ± 9.39	0.17
BMI (kg/m^2^)	27.50 ± 3.24	27.24 ± 3.11	0.93
Systolic blood pressure (mmHg)	128.34 ± 1.45	128.56 ± 1.52	0.86
Diastolic blood pressure (mmHg)	74.56 ± 1.27	74.32 ± 1.51	0.81
TNF (pg/mL)	4.63 ± 1.61	4.74 ± 1.44	0.73
IL-1β (pg/mL)	0.42 ± 0.26	0.47 ± 0.24	0.31
IL-6 (pg/mL)	8.75 ± 1.10	8.94 ± 10.56	0.48
Glutathione peroxidase (U/L)	6050 ± 2045.6	6596 ± 2898.47	0.43
Superoxide dismutase (U/mL)	241.33 ± 55.11	231.87 ± 37.14	0.58
Malondialdehyde (μmol/L)	2.01 ± 0.49	1.98 ± 0.45	0.84
Total body fat (%)	25.27 ± 6.58	24.90 ± 7.35	0.72
Waist circumference (cm)	95.33 ± 9.07	94.49 ± 10.16	0.60
Non-esterified fatty acid (μmol/L)	0.92 ± 0.46	1.01 ± 0.39	0.30
Liver stiffness (kPa)	7.29 ± 3.67	6.65 ± 2.41	0.77
CAP (db/m)	281.97 ± 29.04	273.67 ± 21.51	0.17
Fatty liver index	59.38 ± 24.21	59.86 ± 24.86	0.16
Hepatic steatosis index	36.73 ± 4.71	36.46 ± 4.45	0.85
Lipid accumulation product	60.07 ± 29.55	57.47 ± 35.88	0.63
Creatinine (mg/dL)	0.82 ± 0.04	0.85 ± 0.03	0.75
AST (U/L)	25.31 ± 0.78	25.34 ± 0.74	0.58
ALT (U/L)	26.78 ± 1.48	28.09 ± 1.35	0.08
History of cerebrovascular disease ^††^	2(5.1%)	1(2.6%)	1.00 ^†^
History of coronary artery disease ^††^	3(7.7%)	3(7.7%)	1.00 ^†^
History of hypertension ^††^	27(69.2%)	24(61.5%)	0.63 ^†^
History of diabetic nephropathy ^††^	5(12.8%)	7(17.9%)	0.75 ^†^
History of dyslipidemia ^††^	28(71.8%)	29(74.4%)	1.00 ^†^
Antihypertensive medications ^††^			
Angiotensin receptor blockers	27(69.2)	29(74.4)	0.80
Calcium channel blockers	9(23.1)	6(15.4)	0.57
Beta blockers	7(17.9)	6(15.4)	1.00
Antidyslipedemic medications ^††^			
Statins	21(53.8)	17(43.6)	0.50

* Data were evaluated by the Mann–Whitney *U* test, except for sex (M:F ratio); ^†^ Chi-square test; ^††^ Values expressed as number (percentage). ALT = alanine transaminase; AST = aspartate aminotransferase; BMI = body mass index; CAP = controlled attenuation parameter; IL-1β = Interleukin-1 beta; IL-6 = Interleukin-6; M:F = Male to Female; TNF = tumor necrosis factor.

**Table 2 nutrients-17-01972-t002:** Comparison of body composition, biochemical markers, and hepatic parameters between groups.

Outcomes	Follow-Up Period (mo)	Placebo		Curcumin		*p* Values *
		Median (IQR)	Min–Max	Median (IQR)	Min–Max	
TNF (pg/mL)	0	5.28 (3.52)	2.64–7.04	5.28 (3.08)	2.64–7.04	NS
	3	6.16 (2.20)	2.64–7.04	4.46 (2.64)	2.64–6.16	<0.001
	6	6.33 (2.71)	2.18–14.26	4.01 (2.50)	2.13–6.60	<0.001
	9	6.69 (2.53)	3.30–14.36	3.99 (2.55)	2.10–6.55	<0.001
	12	7.06 (3.55)	2.75–15.37	3.28 (1.58)	1.35–6.55	<0.001
IL-1β (pg/mL)	0	0.43 (0.44)	0.03–0.86	0.43 (0.43)	0.02–0.88	NS
	3	0.51 (0.46)	0.02–0.76	0.38(0.40)	0.02–0.84	0.021
	6	0.89 (0.27)	0.20–1.44	0.42 (0.26)	0.12–0.89	<0.001
	9	0.92 (0.18)	0.20–1.45	0.42 (0.19)	0.13–0.64	<0.001
	12	0.93 (0.17)	0.32–1.46	0.31 (0.18)	0.12–0.54	<0.001
IL-6 (pg/mL)	0	8.80 (1.76)	7.04–10.55	8.80 (1.76)	7.04–10.56	NS
	3	9.27 (2.64)	7.04–10.56	8.28 (2.64)	7.04–10.56	0.002
	6	12.99 (4.50)	7.55–17.99	8.49 (4.72)	3.50–12.40	<0.001
	9	13.24 (3.35)	7.65–18.00	7.60 (5.57)	3.21–13.24	<0.001
	12	14.39 (3.85)	4.33–18.50	6.35 (5.44)	3.10–12.40	<0.001
GPx (U/L)	0	6243 (2549)	1083–9220	6717 (2939)	1124–11,969	NS
	3	6044 (2938)	3540–19,484	6591 (3303)	4252–13,425	<0.001
	6	6189 (1643)	2550–18,386	7987 (2807)	4549–17,679	<0.001
	9	5367 (1817)	3769–8965	9898 (2143)	6089–15,436	<0.001
	12	4653 (1231)	3576–7980	12,468 (3586)	5874–17,790	<0.001
SOD (U/mL)	0	231 (76)	153–379	218 (50)	192–348	NS
	3	214 (79)	153–420	236 (43)	192–348	0.027
	6	211 (58)	150–420	257 (63)	195–356	0.01
	9	206 (20)	150–246	269 (63)	243–356	<0.001
	12	180 (30)	120–211	310 (58.5)	269–399	<0.001
MDA (μmol/L)	0	1.99 (0.73)	1.20–3.33	1.90 (0.58)	1.30–3.23	NS
	3	2.19 (0.52)	1.21–3.42	1.92 (0.77)	1.12–3.17	0.027
	6	2.28 (0.47)	0.77–3.54	1.87 (0.82)	1.22–2.64	<0.001
	9	2.32 (0.590	1.23–3.90	1.70 (0.65)	0.93–2.50	<0.001
	12	2.39 (0.83)	1.20–3.96	1.35 (0.43)	0.93–2.20	<0.001
TBF (%)	0	24.57(9.55)	13.77–36.89	23.16(9.64)	11.98–40.77	NS
	3	25.71(8.08)	13.03–38.54	21.53(9.18)	12.19–39.10	NS
	6	25.88(7.88)	14.91–39.14	20.76(9.16)	11.76–41.16	0.001
	9	26.77(8.95)	15.75–39.74	21.20(8.26)	11.39–39.62	0.003
	12	27.08(8.93)	16.48–42.82	20.19(7.84	11.25–36.90	0.001
WC (cm)	0	95 (11)	76.00–112.00	95 (12)	74.00–120.00	NS
	3	98 (10)	75.00–113.00	94 (9)	75.00–117.00	NS
	6	100 (9.5)	78.00–115.00	93 (9)	75.00–135.00	0.01
	9	99 (11)	78.00–117.00	92 (10)	74.00–117.00	0.003
	12	99 (10.5)	76.00–118.00	91 (11)	73.00–114.00	<0.001
NEFA (μmol/L)	0	0.85 (0.57)	0.20–1.98	0.84 (0.5)	0.30–1.73	NS
	3	1.15 (0.66)	0.34–1.86	1.14 (0.57)	0.34–1.79	NS
	6	1.18 (0.78)	0.32–1.89	0.87 (0.58)	0.34–1.85	0.001
	9	1.19 (0.58)	0.45–1.98	0.87 (0.48)	0.34–1.79	0.007
	12	1.25 (0.75)	0.40–1.95	0.88 (0.53)	0.20–1.92	0.002
BMI (kg/m^2^)	0	27.06 (4.79)	19.05–34.77	27.68 (4.87)	20.43–36.58	NS
	3	27.63 (4.25)	17.89–33.69	27.10 (4.45)	19.22–36.20	NS
	6	27.78 (3.68)	18.07–33.46	26.33 (4.45)	18.80–35.16	0.031
	9	27.46 (4.77)	17.89–33.03	26.08 (4.29)	19.22–36.72	0.017
	12	27.70 (4.16)	17.72–33.32	25.97 (4.05)	20.55–35.55	0.002
Glucose (mg/dL)	0	126 (26.5)	91–181	122 (22)	79–163	NS
	3	128 (36.5)	102–195	124 (28)	80–171	NS
	6	131 (37.5)	98–214	123 (27)	79–171	0.029
	9	130 (27.0)	105–185	120 (18)	75–150	<0.001
	12	131 (26.5)	98–187	117 (20)	70–151	<0.001
HbA1C (%)	0	6.2 (0.85)	5.1–8.9	6.3 (0.70)	4.8–7.8	NS
	3	6.6 (0.70)	5.4–8.9	6.3 (0.70)	4.8–8.0	NS
	6	6.6 (1.00)	5.4–9.0	6.2 (1.05)	4.5–8.3	0.030
	9	6.7 (1.10)	5.6–10.3	6.1 (1.05)	4.3–8.2	0.003
	12	6.6 (1.20)	5.3–9.9	5.9 (0.90)	4.2–8.4	0.003
Liver stiffness (kPa)	0	6.2 (3.60)	3.20–19.60	6.6 (3.15)	2.80–14.20	NS
	3	6.6 (2.70)	3.90–14.30	6.0 (2.00)	3.00–9.50	<0.001
	6	7.2 (2.15)	4.20–12.10	5.9 (2.16)	3.00–9.46	<0.001
	9	7.9 (3.20)	4.80–13.50	5.5 (1.90)	3.00–8.11	<0.001
	12	6.9 (3.45)	5.00–13.50	4.2 (1.60)	2.90–7.60	<0.001
CAP (dB/m)	0	279 (34.5)	248–387	264 (30)	248.00–323	NS
	3	258 (49)	184–365	287 (34)	147.00–319	<0.001
	6	287 (34)	203–353	263 (53)	100.00–305	<0.001
	9	280 (67.5)	202–293	250 (58)	102.00–293	<0.001
	12	292 (64.5)	199–35	223 (51)	107.00–274	<0.001
FLI	0	64.09 (35.29)	12.68–92.39	60.84 (35.35)	10.64–98.71	NS
	3	66.37 (31.92)	14.42–92.41	57.89 (39.49)	13.99–98.66	0.035
	6	69.92 (32.71)	16.64–95.36	46.64(27.37)	10.21–98.86	0.001
	9	71.45 (40.01)	15.77–96.14	45.46(31.39)	10.82–98.35	0.002
	12	76.43 (36.25)	17.36–97.55	39.64 (35.92)	9.21–97.44	<0.001
HSI	0	36.50 (7.28)	28.39–47.15	36.46 (7.19)	28.39–43.63	NS
	3	37.91 (6.43)	29.26–50.37	36.62 (5.41)	27.11–48.41	0.042
	6	38.15 (6.03)	27.95–49.92	36.49 (5.27)	26.58–47.82	0.033
	9	38.30 (5.67)	29.26–52.91	36.22 (5.39)	26.02–47.46	0.033
	12	38.95 (6.09)	29.99–54.44	35.92 (5.61)	23.57–48.38	0.010
LAP	0	53.68 (34.94)	12.61–139.17	45.86 (31.25)	13.01–176.37	NS
	3	64.38 (39.28)	25.32–128.47	44.73 (31.45)	12.87–173.44	0.003
	6	72.86 (42.36)	29.14–184.22	43.38 (27.71)	13.42–148.20	0.003
	9	59.78 (45.73)	29.62–207.00	40.72 (19.80)	12.04–144.61	0.004
	12	65.24 (53.58)	30.90–191.59	39.85 (23.38)	12.19–136.90	<0.001

* *p* values were evaluated by the Mann–Whitney *U* test. BMI = body mass index; CAP = controlled attenuated parameter; GPx = glutathione peroxidase; FLI = fatty liver index; HbA1C = glycated hemoglobin; HSI = hepatic steatosis Index; IL-1β = Interleukin-1 beta; IL-6 = interleukin-6; IQR = Interquartile range; LAP = lipid accumulation product; MDA = malondialdehyde; mo = months; NEFA = non-esterified fatty acid; NS = not significant; SOD = superoxide dismutase; TBF = total body fat; TNF = tumor necrosis factor; WC = Waist circumference

## Data Availability

The original contributions presented in the study are included in the article, further inquiries can be directed to the corresponding author.

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
