# Peer review of "Curcumin for Inflammation Control in Individuals with Type 2 Diabetes Mellitus and Metabolic Dysfunction-Associated Steatotic Liver Disease: A Randomized Controlled Trial"

_nutrients, 2025, doi:10.3390/nu17121972_

Round 1
Reviewer 1 Report
Comments and Suggestions for Authors
This manuscript by Yaikwawong and coworkers, presents a well-conducted randomized, double-blind, placebo-controlled clinical trial assessing the efficacy of curcumin supplementation in patients with metabolic dysfunction-associated steatotic liver disease (MASLD). The study addresses a highly relevant and timely topic, considering the increasing prevalence of MASLD and the limited pharmacological interventions currently available. The trial’s design, execution, and analysis are generally sound and contribute valuable data to the field of nutritional and metabolic research.
Major comments:
- While non-invasive measures like FibroScan and CAP score are increasingly used, the lack of liver biopsy data limits conclusions regarding histological improvements (e.g., inflammation, ballooning, fibrosis stages). The authors should acknowledge this limitation more explicitly in the discussion.
- The sample size calculation is based solely on changes in TNF-α levels. Although TNF-α is the primary endpoint, a power analysis for secondary outcomes (e.g., liver stiffness or CAP score) would further support the robustness of the findings.
- While blinding procedures are described, the success of blinding (e.g., through post-intervention survey questions) is not discussed. Adding information on whether patients or investigators could guess their allocation would improve transparency.
- The manuscript notes a predominance of female participants, which may affect generalizability, particularly as MASLD prevalence differs by sex. A stratified analysis or sensitivity test by gender could help address this.
- The authors mention diet questionnaires, but results from these assessments are not presented. Given the importance of lifestyle factors in MASLD progression, changes in dietary intake should be described to ensure comparability between groups.
Minor Comments:
- Line 273 (Results section): “NEFA levels were significantly lower…” the authors could briefly explain the pathophysiological significance of NEFAs in MASLD.
- The manuscript uses both “CAP score” and “controlled attenuation parameter.” Standardize this throughout.
- Ensure all abbreviations are defined in each table caption for clarity.
- In several places, there are minor typographical issues (e.g., formatting of statistical values, punctuation), which should be corrected in final proofreading.
Author Response
Reviewer 1
Major comments
Comments 1
While non-invasive measures like FibroScan and CAP score are increasingly used, the lack of liver biopsy data limits conclusions regarding histological improvements (e.g., inflammation, ballooning, fibrosis stages). The authors should acknowledge this limitation more explicitly in the discussion.
Response:
Thank you for your suggestion. We have now explicitly acknowledged this in the revised manuscript (line 472-476).
“First, MASLD was not histologically confirmed, as liver biopsies were not performed. While non-invasive methods like FibroScan provide quantitative measures of liver stiffness and steatosis, respectively, they may not fully capture subtle histological changes, potentially leading to misclassification, especially in cases where such changes are minimal yet clinically significant 1.
Comments 2
The sample size calculation is based solely on changes in TNF-α levels. Although TNF-α is the primary endpoint, a power analysis for secondary outcomes (e.g., liver stiffness or CAP score) would further support the robustness of the findings.
Response:
Thank you for your suggestion. We agree that additional power calculations for secondary outcomes, such as liver stiffness measurement in the method section in the manuscript accordingly (line 252-257).
“While TNF is the primary endpoint, secondary outcome such as liver stiffness is also critical for evaluating liver fibrosis. Based on prior studies evaluating changes in liver stiffness, a sample size of 34 participants was estimated to provide 90% power to detect a 3.1 kPa reduction in liver stiffness 2. This underscores the necessity of tailoring sample size calculations to the specific effect sizes and variability expected in secondary outcome.”
Comment 3:
While blinding procedures are described, the success of blinding (e.g., through post-intervention survey questions) is not discussed. Adding information on whether patients or investigators could guess their allocation would improve transparency.
Response:
Thank you for your suggestion. We have now included a statement in the Methods and result sections indicating that post-intervention surveys were conducted.
Methods (line 158-162)
“To evaluate the effectiveness of blinding, both participants and investigators were asked to guess the assigned intervention group (curcumin or placebo) at the conclusion of the study. The accuracy of these guesses was then compared to the expected probability of correct guesses under random chance (50%) using a two-sided exact binomial test.”
Results (line 334-340)
“Blinding assessment Both participants and investigators were asked to guess the assigned treatment group at the end of the study. As shown in the Supplemental Table 3, the accuracy of these guesses did not significantly differ from random chance. Among participants, 48.7% correctly identified their group allocation (p = 0.84), and among investigators, 52.6% guessed correctly (p = 0.73). These results suggest that blinding was effectively preserved throughout the trial.”
Comments 4:
The manuscript notes a predominance of female participants, which may affect generalizability, particularly as MASLD prevalence differs by sex. A stratified analysis or sensitivity test by gender could help address this.
Response:
Thank you for this valuable comment. We include a sex-stratified analysis in the methods, results and discussion section.
“Methods (line 267-270)
To assess potential sex-specific responses to curcumin supplementation, we conducted sex-stratified linear regression analyses on the change (Δ) in key outcomes over the 12-month intervention period. Each model was adjusted for baseline values and age to control for potential confounding factors.
Results (line 329-333)
Sex-stratified analyses Curcumin supplementation significantly improved inflammatory markers,
oxidative stress, anthropometric indices, metabolic profiles, and hepatic parameters in both male and female participants. Comprehensive results are presented in the Supplemental Table 2.
Discussion (line 462-468)
“Despite the female-dominant cohort, sex-stratified analyses revealed that curcumin supplementation consistently improved inflammatory markers, oxidative stress, anthropo-metric measures, metabolic parameters, and hepatic steatosis in both sexes. These effects remained significant after adjusting for age, suggesting that curcumin’s therapeutic bene-fits are not sex-dependent. Nonetheless, future studies with larger, more sex-balanced populations are needed to validate these findings and explore potential sex-specific responses.”
Comment 5:
The authors mention diet questionnaires, but results from these assessments are not presented. Given the importance of lifestyle factors in MASLD progression, changes in dietary intake should be described to ensure comparability between groups.
Response:
Thank you for your comment. we have included a summary of dietary intake changes in the revised manuscript to ensure comparability between groups in result section (line 281-288).
“Additionally, Supplemental Table 2 summarizes the mean daily nutrient in-take—covering total energy intake (kcal/day), macronutrient distribution (percentage of energy from carbohydrates, protein, and fat), and fiber intake (g/day)—assessed at baseline and after 12 months using validated dietary questionnaires. No significant be-tween-group differences were observed at either time point, indicating that dietary intake remained stable throughout the study. These findings support the comparability of life-style factors between groups and reduce the likelihood that dietary variation confounded the observed effects of the intervention.”
Minor comments
Comments 1:
Line 273 (Results section): “NEFA levels were significantly lower…” the authors could briefly explain the pathophysiological significance of NEFAs in MASLD.
Response:
Thank you for your suggestion: We revised the text in the result section in the revised manuscript accordingly (line 306-309).
“Non-esterified fatty acid levels
Curcumin significantly reduced NEFA levels—a key driver of hepatic fat accumulation and MASLD pathogenesis—compared to placebo at 6 months (Table 2), suggesting enhanced liver metabolic function and reduced lipotoxicity.”
Comments 2:
The manuscript uses both “CAP score” and “controlled attenuation parameter.” Standardize this throughout.
Response:
We have standardized the terminology throughout the manuscript to CAP.
“Methods
Participants were eligible if they were at least 35 years old and had hepatic steatosis confirmed by FibroScan (Echosens, Paris, France) with a controlled attenuation parameter (CAP) greater than 248 dB/m 3 (line 90-92).
Regular monitoring and verification were conducted to confirm the maintenance of blinding and to ensure consistent CAP and liver stiffness (line 144-145).”
The secondary outcomes included changes in CAP and liver stiffness, measured noninvasively by transient elastography (FibroScan 520 Touch; Echosens, Paris, France) (line 181-183).
“These steps enhanced the reproducibility of CAP and liver stiffness assessments (line 189-190).”
Results
CAP and liver stiffness
The CAP was significantly lower in the curcumin group than in the placebo group at 3, 6, 9, and 12 months (Table 2) (line 319-322).
Discussion
This double-blind, placebo-controlled trial used the FibroScan 520 Touch, a noninvasive device that measures hepatic steatosis via the CAP 4 (line 365-366).
Curcumin’s anti-inflammatory and antioxidant properties may underlie its beneficial metabolic effects, including reductions in waist circumference, TBF, visceral fat, and BMI. These changes could explain the observed decreases in CAP. In our study, 12 months of curcumin supplementation significantly lowered CAP (line 432-435).”
Comments 2:
Ensure all abbreviations are defined in each table caption for clarity.
Response:
Thank you for valuable suggestion. All abbreviations have now been defined in the captions of the relevant tables.
“Table 1
Abbreviations (line 111-113)
ALT=alanine transaminase; AST=aspartate aminotransferase; BMI=body mass index; CAP = con-trolled attenuation parameter; IL-1β = Interleukin-1 beta; IL-6 = Interleukin-6; M:F=Male to Female; TNF=tumor necrosis factor.
Table 2
Abbreviations (line 295-299)
BMI=body mass index; CAP=controlled attenuated parameter; GPx=glutathione peroxidase; FLI=Fatty liver index; HbA1C= glycated hemoglobin; HSI=Hepatic Steatosis Index; IL-1β = Interleukin-1 beta; IL-6=interleukin-6; IQR Interquartile range; LAP=Lipid accumulation product; MDA=malondialdehyde; NEFA=non-esterified fatty acid SOD=superoxide dismutase; TBF= Total Body fat; TNF=tumor necrosis factor; WC = Waist circumference”
Comments 3:
In several places, there are minor typographical issues (e.g., formatting of statistical values, punctuation), which should be corrected in final proofreading.
Response:
Thank you for suggestion. Formatting issues errors have been corrected.
Abstract (line 27-31)
Results All participants completed the study (curcumin group: n = 39; placebo group: n = 39). Curcumin significantly reduced TNF levels at all follow-up points compared to placebo (p < 0.001). IL-1β, IL-6, and malondialdehyde levels also declined significantly (p < 0.001), while antioxidant enzyme activities, including glutathione peroxidase and superoxide dismutase, increased significantly (p < 0.001), indicating improved oxidative balance.
Methods (line 211)
TBF = (BFP × total body weight [kg])/100

Reviewer 2 Report
Comments and Suggestions for Authors
The study „Curcumin for Inflammation Control in Metabolic Dysfunction-Associated Steatotic Liver Disease: A Randomized Controlled Trial” submitted by Metha Yaikwawong et al. showed that a one year supplementation with curcumin improved inflammation in patients with MASLD.
This is a well-performed trial with a detailed description of the patients.
Introduction:“Despite these promising findings, the number of randomized clinical trials on curcumin supplementation remains limited.” The authors should provide a more detailed description of some of the previous studies. They should emphasise that most of these studies were much shorter than theirs.
Authors have to include in the title and the abstract that all patients were diabetic.
Figure S2, please correct “145 excluded due to no evidence of liver”
Details of all ELISAs used should be included in Data collection and measurement methods
GPT/GOT and AST/ALT are used. Please use only one abbreviation.
TNF-alpha should be TNF.
Which programs were used to determine the sample size and the normal distribution of the data?
“Chronic low-grade inflammation is associated with obesity and metabolic disorders, and curcumin’s inhibition of inflammatory pathways may support weight loss” Inflammation is usually linked to increased energy needs and weight loss. This explanation given by the authors is unlikely. Please correct.
The data regarding liver stiffness needs to be better discussed. Liver stiffness improves with reduced inflammation and lower kPA in the patients may indicate less inflammation rather than less liver stiffness.
Conclusions, please use upper case letter for curcumin.
Author Response
Reviewer 2
Comments 1
“Despite these promising findings, the number of randomized clinical trials on curcumin supplementation remains limited.” The authors should provide a more detailed description of some of the previous studies. They should emphasise that most of these studies were much shorter than theirs.”
Response:
Thank you for your suggestion. We have revised the introduction to provide more detail on previous trials in the revised manuscript accordingly (line 65-74).
Response:
“Several randomized clinical trials have demonstrated that curcumin possesses hepatoprotective, antioxidant, anti-inflammatory, antidiabetic, and lipid-lowering properties in humans. Notably, most of these studies were of relatively short duration, typically around 8 weeks. For instance, Rahmani et al. (2016) conducted an 8-week randomized, double-blind, placebo-controlled trial in patients with non-alcoholic fatty liver disease (NAFLD), finding that curcumin supplementation significantly reduced liver fat content and improved metabolic parameters compared to placebo 5. Similarly, Panahi et al. (2017) reported that 8 weeks of phytosomal curcumin supplementation led to significant improvements in liver enzymes and ultrasonographic findings in NAFLD patients 6. These findings suggest that curcumin may be beneficial in managing NAFLD.”
Comments 2
“Authors have to include in the title and the abstract that all patients were diabetic.”
Response:
Thank you for your suggestion. We have amended both the title and abstract to specify that all participants had type 2 diabetes mellitus.
“Title
Curcumin for Inflammation Control in Individuals with Type 2 Diabetes Mellitus and Metabolic Dysfunction-Associated Steatotic Liver Disease: A Randomized Controlled Trial (line 1-3)
Abstract:
In this randomized, double-blind, placebo-controlled trial, 78 patients with type 2 diabetes mellitus (T2DM) and MASLD were randomly assigned to receive either curcumin (1500 mg/day) or placebo for 12 months (line 20-22).
Comment 3
“Figure S2, please correct “145 excluded due to no evidence of liver”
Response:
Thank you for your suggestion. We have corrected to “145 excluded due to no evidence of liver steatosis in the Supplementary FIGURE 2.
Comments 3
Details of all ELISAs used should be included in Data collection and measurement methods
Response:
Thank you for suggestion. We have included specific catalog numbers and performance characteristics of the ELISA kits used in the “Data collection and measurement methods” section (line 217-227).
“Serum levels of pro-inflammatory cytokines—interleukin-1 beta (IL-1β), interleukin-6 (IL-6), and tumor necrosis factor (TNF)—were successfully measured using validated enzyme-linked immunosorbent assays (ELISAs). Blood samples were collected, processed, and stored according to standard protocols to ensure sample integrity. IL-1β was measured using the Human IL-1β ELISA Kit (Abcam, catalog no. ab214025), which has a sensitivity of 14.06 pg/mL and intra- and inter-assay coefficients of variation (CVs) of 4.8% and 5.6%, respectively. IL-6 was assessed using the Human IL-6 ELISA Kit (Abcam, catalog no. ab178013), with a sensitivity of 2 pg/mL and CVs of 4.2% (intra-assay) and 6.4% (inter-assay). TNF levels were determined using the Human TNF ELISA Kit (Abcam, catalog no. ab181421), with a sensitivity of 4.32 pg/mL and intra- and inter-assay CVs of 2.5% and 3.1%, respectively.”
Comments 4:
GPT/GOT and AST/ALT are used. Please use only one abbreviation.
Response:
Thank you for suggestion: We have standardized all enzyme abbreviations to AST and ALT throughout the manuscript for consistency (line 243-244).
“Hepatic steatosis index (HSI) was computed using this formula 7:
HSI = 8 × (ALT/AST ratio) + BMI (+ 2 for women).
Comments 5:
TNF-alpha should be TNF.
Response:
Thank you for suggestion: We have TNF-alpha” with “TNF” throughout the manuscript.
“Abstract
The primary outcome was the change in tumor necrosis factor (TNF) levels (line 22-23).
All participants completed the study (curcumin group: n = 39; placebo group: n = 39). Curcumin significantly reduced TNF levels at all follow-up points compared to placebo (p < 0.001) (line 27-28).
Methods
The primary outcome was a significant reduction in TNF levels in the curcumin group compared with the placebo group (line 180-181).
Serum levels of pro-inflammatory cytokines—interleukin-1 beta (IL-1β), interleukin-6 (IL-6), and tumor necrosis factor (TNF)—were successfully measured using validated enzyme-linked immunosorbent assays (ELISAs) (line 217-219).
TNF levels were determined using the Human TNF ELISA Kit (Abcam, catalog no. ab181421), with a sensitivity of 4.32 pg/mL and intra- and inter-assay CVs of 2.5% and 3.1%, respectively (line 225-227).
The initial sample size calculation was based on detecting a 2 pg/mL difference in mean serum TNF levels, with 80% statistical power (β = 0.20), requiring 21 participants per group 8. While TNF is the primary endpoint, secondary outcome such as liver stiffness is also critical for evaluating liver fibrosis (line 251-253). While TNF is the primary endpoint, secondary outcome such as liver stiffness is also critical for evaluating liver fibrosis (line 250-253).
Results
Anti-inflammatory effects
Compared with placebo, curcumin significantly reduced pro-inflammatory cytokines—including TNF, IL-1β, and IL-6—at 3, 6, 9, and 12 months (Table 2) (line 290-291).
Discussion
Our findings revealed that 3 months of curcumin supplementation significantly decreased pro-inflammatory cytokines—TNF, IL-1β, and IL-6—compared with placebo (line 366-368).
Studies show that curcumin reduces IL-1β 9, IL-6 10, and TNF 10, key cytokines involved in the pathogenesis of MASLD (line 381-382).
In our study, a 3-month curcumin intervention significantly reduced TNF, IL-1β, and IL-6 levels in patients with MASLD. These results align with those reported by Saadatiet al, who observed lower TNF levels after a 12-week curcumin supplementation in patients with non-alcoholic fatty liver disease 11. Jazayeri-Tehrani et al similarly noted decreased TNF and IL-6 levels following 3 months of nanocurcumin administration 12, while Ghaffari et al documented reduced IL-6 after a 10-week turmeric regimen 13 (line 388-393).
Tables
Table 1. Baseline characteristics of study participants.
|
TNF (pg/ml) |
4.63±1.61 |
4.74±1.44 |
0.73 |
Abbreviation (line 113)
TNF=tumor necrosis factor
Table 2. Comparison of body composition, biochemical markers, hepatic parameters between groups.
|
TNF (pg/ml) |
0 |
5.28 (3.52) |
2.64-7.04 |
5.28 (3.08) |
2.64-7.04 |
NS |
|
|
3 |
6.16 (2.20) |
2.64-7.04 |
4.46 (2.64) |
2.64-6.16 |
<0.001 |
|
|
6 |
6.33 (2.71) |
2.18-14.26 |
4.01 (2.50) |
2.13-6.60 |
<0.001 |
|
|
9 |
6.69 (2.53) |
3.30-14.36 |
3.99 (2.55) |
2.10-6.55 |
<0.001 |
|
|
12 |
7.06 (3.55) |
2.75-15.37 |
3.28 (1.58) |
1.35-6.55 |
<0.001 |
Abbreviation (line 299)
TNF=tumor necrosis factor
Abbreviations used in the text of the manuscript (line 508)
TNF: tumor necrosis factor
Comments 6:
Which programs were used to determine the sample size and the normal distribution of the data?
Response:
Thank you for your suggestion. We revised the method section to include the following details:
“Sample size (line 257-258).
Sample size calculations were performed using G*Power software (version 3.1) 14 (
Statistical analysis (line 271-272)
The assessment of normality and all other statistical analyses were conducted using R software version 4.3.2 (R Foundation for Statistical Computing, Vienna, Austria) 15.”
Comments 7:
Chronic low-grade inflammation is associated with obesity and metabolic disorders, and curcumin’s inhibition of inflammatory pathways may support weight loss” Inflammation is usually linked to increased energy needs and weight loss. This explanation given by the authors is unlikely. Please correct.
Response:
Thank you for your suggestion. We have revised the discussion in the revised manuscript accordingly (line 416-423).
While inflammation is typically associated with increased energy expenditure and weight loss in conditions like cachexia, obesity is characterized by chronic low-grade inflammation that contributes to insulin resistance and metabolic dysregulation, thereby hindering weight loss 16 17. Curcumin's anti-inflammatory properties may help restore metabolic balance by improving insulin sensitivity, modulating adipokines, and regulating lipid metabolism 18. Clinical trial has demonstrated that curcumin supplementation can reduce body weight and BMI, even in settings with controlled lifestyle interventions 18,19.
Comments 8
Thank you for your suggestions. We have clarified the discussion in the revise manuscript accordingly (line 441-448)
The data regarding liver stiffness needs to be better discussed. Liver stiffness improves with reduced inflammation and lower kPA in the patients may indicate less inflammation rather than less liver stiffness.
Response
“We observed significant reductions in liver stiffness. Although liver stiffness is a widely used non-invasive marker of hepatic fibrosis, it is also sensitive to hepatic inflammation. Therefore, elevated liver stiffness may reflect inflammatory activity rather than true fibrotic changes, and the observed reductions may primarily indicate improvements in inflammation rather than definitive fibrosis regression 20. Curcumin’s well-established anti-inflammatory properties likely contributed to these changes 21. However, as liver stiffness is influenced by both fibrosis and inflammation, the precise nature of the improvements remains uncertain without histological confirmation.”
Comments 9
Conclusions, please use upper case letter for curcumin.
Response:
Thank you for your suggestion. We have corrected the conclusion in the revise manuscript accordingly (line 481-482).
“Curcumin, a bioactive compound in turmeric, exhibits promising hepatoprotective effects through its anti-inflammatory and antioxidant properties.”
